# Reinforcement and Imitation Learning for Diverse Visuomotor Skills

## Abstract

We propose a general deep reinforcement learning method and apply it to robot manipulation tasks. Our approach leverages demonstration data to assist a reinforcement learning agent in learning to solve a wide range of tasks, mainly previously unsolved. We train visuomotor policies end-to-end to learn a direct mapping from RGB camera inputs to joint velocities. Our experiments indicate that our reinforcement and imitation approach can solve contact-rich robot manipulation tasks that neither the state-of-the-art reinforcement nor imitation learning method can solve alone. We also illustrate that these policies achieved zero-shot sim2real transfer by training with large visual and dynamics variations.

## 1 Introduction

Recent advances in deep reinforcement learning (RL) have performed very well in several challenging domains such as video games (Mnih et al., 2015) and Go (Silver et al., 2016). For robotics, RL in combination with powerful function approximators provides a general framework for designing sophisticated controllers that would be hard to handcraft otherwise. Yet, despite significant leaps in other domains the application of deep RL to control and robotic manipulation has proven challenging. While there have been successful demonstrations of deep RL for manipulation (e.g. Nair et al. 2017; Popov et al. 2017) and also noteworthy applications on real robotic hardware (e.g. Levine et al. 2015; Yahya et al. 2016) there have been very few examples of learned controllers for sophisticated tasks even in simulation.

Robotics exhibits several unique challenges. These include the need to rely on multi-modal and partial observations from noisy sensors, such as cameras. At the same time, realistic tasks often come with a large degree of variation (visual appearance, position, shapes, etc.) posing significant generalization challenges. Training on real robotics hardware can be daunting due to constraints on the amount of training data that can be collected in reasonable time. This is typically much less than the millions of frames needed by modern algorithms. Safety considerations also play an important role, as well as the difficulty of accessing information about the state of the environment (like the position of an object) e.g. to define a reward. Even in simulation when perfect state information and large amounts of training data are available, exploration can be a significant challenge. This is partly due to the often high-dimensional and continuous action space, but also due to the difficulty of designing suitable reward functions.

In this paper, we present a general deep reinforcement learning method that addresses these issues and that can solve a wide range of robot arm manipulation tasks directly from pixels, most of which have not been solved previously. Our key insight is 1) to reduce the difficulty of exploration in continuous domains by leveraging a handful of human demonstrations; 2) several techniques to stabilize the learning of complex manipulation policies from vision; and 3) to improve generalization by increasing the diversity of the training conditions. As a result, the trained policies work well under significant variations of system dynamics, object appearances, task lengths, etc. We ground these policies in the real world, demonstrating zero-shot transfer from simulation to real hardware.

We develop a new method to combine imitation learning with reinforcement learning. Our method requires only a small number of human demonstrations to dramatically simplify the exploration problem. It uses demonstration data in two ways: first, it uses a hybrid reward that combines sparse environment reward with imitation reward based on Generative Adversarial Imitation Learning (Ho

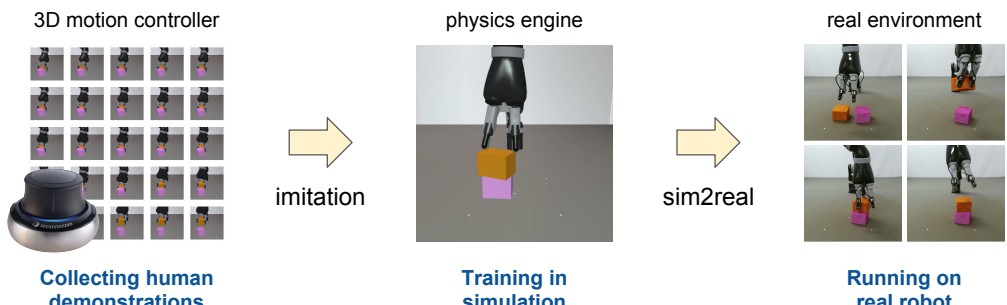

3D motion controller  physics engine  real environment

imitation  sim2real

**Collecting human demonstrations**  **Training in simulation**  **Running on real robot**

Figure 1: Our proposal of a principled robot learning pipeline. We used 3D motion controllers to collect human demonstrations of a task. Our reinforcement and imitation learning model leveraged these demonstrations to facilitate learning in a simulated physical engine. We then performed sim2real transfer to deploy the learned visuomotor policy to a real robot.

& Ermon, 2016), which produces more robust controllers; second, it uses demonstration as a curriculum to initiate training episodes along demonstration trajectories, which facilitates the agent to reach new states and solve longer tasks. As a result, it solves dexterous manipulation tasks that neither the state-of-the-art reinforcement learning nor imitation learning method can solve alone.

Previous RL-based robot manipulation policies (Nair et al., 2017; Popov et al., 2017) largely rely on low-level states as input, or use severely limited action spaces that ignore the arm and instead learn Cartesian control of a simple gripper. This limits the ability of these methods to represent and solve more complex tasks (e.g., manipulating arbitrary 3D objects) and to deploy in real environments where the privileged state information is unavailable. Our method learns an end-to-end visuomotor policy that maps RGB camera observations to joint space control over the full 9-DoF arm (6 arm joints plus 3 actuated fingers).

To sidestep the constraints of training on real hardware we embrace the sim2real paradigm which has recently shown promising results (James et al., 2017; Rusu et al., 2016a). Through the use of a physics engine and high-throughput RL algorithms, we can simulate parallel copies of a robot arm to perform millions of complex physical interactions in a contact-rich environment while eliminating the practical concerns of robot safety and system reset. Furthermore, we can, during training, exploit privileged information about the true system state with several new techniques, including learning policy and value in separate modalities, an object-centric GAIL discriminator, and auxiliary tasks for visual modules. These techniques stabilize and speed up policy learning from pixels.

Finally, we diversify training conditions such as visual appearance as well as e.g. the size and shape of objects. This improves both generalization with respect to different task conditions as well as transfer from simulation to reality.

To demonstrate our method, we use the same model and the same algorithm for visuomotor control of six diverse robot arm manipulation tasks. Combining reinforcement and imitation, our policies solve the tasks that the state-of-the-art reinforcement and imitation learning cannot solve and outperform human demonstrations. Our approach sheds light on a principled deep visuomotor learning pipeline illustrated in Fig. 1, from collecting real-world human demonstration to learning in simulation, and back to real-world deployment via sim2real policy transfer.

## 2 RELATED WORK

Reinforcement learning methods have been extensively used with low-dimensional policy representations such as movement primitives to solve a variety of control problems both in simulation and in reality. Three classes of RL algorithms are currently dominant for continuous control problems: guided policy search methods (GPS; Levine & Koltun 2013), value-based methods such as the deterministic policy gradient (DPG; Silver et al. 2014; Lillicrap et al. 2016; Heess et al. 2015) or the normalized advantage function (NAF; Gu et al. 2016b) algorithm, and trust-region based policy gradient algorithms such as trust region policy optimization (TRPO) and proximal policy optimization

(PPO). TRPO (Schulman et al., 2015) and PPO (Schulman et al., 2017) hold appeal due to their robustness to hyper-parameter settings as well as their scalability (Heess et al., 2017) but the lack of sample efficiency makes them unsuitable for training directly on robotics hardware.

GPS (Levine & Koltun, 2013) has been used e.g. by Levine et al. (2015) and Yahya et al. (2016) to learn visuomotor policies directly on a real robotics hardware after a network pretraining phase. Gupta et al. (2016) and Kumar et al. (2016) use GPS for learning controllers for robotic hand models. Value-based methods have been employed, e.g. by Gu et al. (2016a) who use NAF to learn a door opening task directly on a robot while Popov et al. (2017) demonstrate how to solve a stacking problem efficiently using a distributed variant of DPG.

The idea of using large-scale data collection for training visuomotor controllers has been the focus of Levine et al. (2016) and Pinto & Gupta (2015) who train a convolutional network to predict grasp success for diverse sets of objects using a large dataset with 10s or 100s of thousands of grasp attempts collected from multiple robots in a self-supervised setting.

An alternative strategy for dealing with the data demand is to train in simulation and transfer the learned controller to real hardware, or to augment real-world training with synthetic data. Rusu et al. (2016b) learn simple visuomotor policies for a Jaco robot arm and transfer to reality using progressive networks (Rusu et al. 2016a). Viereck et al. (2017) minimize the reality gap by relying on depth. Tobin et al. (2017) use visual variations to learn robust object detectors that can transfer to reality; James et al. (2017) combine randomization with supervised learning. Bousmalis et al. (2017) augments the training with simulated data to learn grasp prediction of diverse shapes.

Suitable cost functions and exploration strategies for control problems are challenging to design, so demonstrations have long played an important role. Demonstrations can be used to initialize policies, design cost functions, guide exploration, augment the training data, or a combination of these. Cost functions can be derived from demonstrations either via tracking objectives (e.g. Gupta et al. 2016) or, via inverse RL (e.g. Boularias et al. 2011; Finn et al. 2016), or, as in our case, via adversarial learning (Ho & Ermon, 2016). When expert actions or expert policies are available, behavioral cloning or DAgger can be used (Rahmatizadeh et al. 2017; James et al. 2017; Duan et al. 2017). Alternatively, expert trajectories can be used as additional training data for off-policy algorithms such as DPG (e.g. Vecerik et al. 2017). Most of these methods require observation and/or action spaces to be aligned between robot and demonstrations. Recently, methods for third person imitation have been proposed (e.g. Sermanet et al. 2017; Liu et al. 2017; Finn et al. 2017).

Concurrently with our work several papers have presented results on manipulation tasks. Rajeswaran et al. (2017); Nair et al. (2017) both use human demonstrations to aid exploration. Nair et al. (2017) extends the DDPGfD algorithm (Vecerik et al., 2017) to learn a block stacking task on a position-controlled arm in simulation. Rajeswaran et al. (2017) use the demonstrations with a form of behavioral cloning and data augmentation to learn several complex manipulation tasks. In both cases, controllers observe a low-dimensional state space representation and the methods inherently require aligned state and action spaces with the demonstrations. Pinto et al. (2017) and Peng et al. (2017) address the transfer from simulation to reality, focusing on randomizing visual appearance and robot dynamics respectively. Peng et al. transfer a block-pushing policy operating from state features to a 7-DoF position controlled Fetch robotics arm. Pinto et al. consider different tasks using visual input with end-effector position control.

## 3  MODEL

Our goal is to learn a deep visuomotor policy for robot manipulation tasks. The policy takes both an RGB camera observation and a proprioceptive feature that describes the joint positions and angular velocities. These two sensory modalities are also available on the real robot, enabling us to perform zero-shot policy transfer once trained in simulation. Fig. 2 provides an overview of our model. The deep visuomotor policy encodes the pixel observation with a convolutional network (CNN) and the proprioceptive feature with a multilayer perceptron (MLP). The features from these two modules are concatenated and passed to a recurrent LSTM layer before producing the joint velocities. The whole network is trained end-to-end. We start with a brief review of the basics of generative adversarial imitation learning (GAIL) and proximal policy optimization (PPO). Our model extends upon these two methods for visuomotor skills.

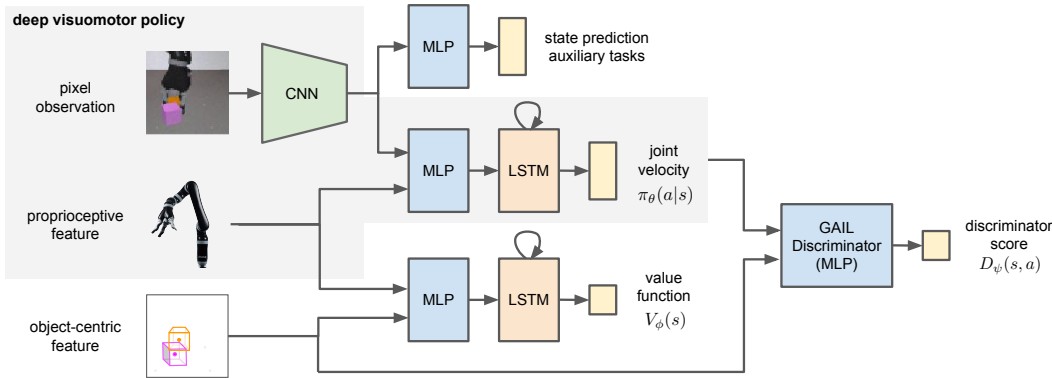

Figure 2: Model overview. The core of our model is the deep visuomotor policy, which takes the camera observation and the proprioceptive feature as input and produces the next joint velocities.

## 3.1 BACKGROUND: GAIL AND PPO

Imitation learning (IL) is the problem of learning a behavior policy by mimicking a set of demonstrations. Here we assume that human demonstration is provided as a dataset of state-action pairs $\mathcal{D} = \{(s_i, a_i)\}_{i=1...N}$. Traditional IL methods cast it as a supervised learning problem, i.e., behavior cloning. These methods use maximum likelihood to train a parameterized policy $\pi_\theta : \mathcal{S} \to \mathcal{A}$, where $\mathcal{S}$ is the state space and $\mathcal{A}$ is the action space, such that $\theta^* = \arg\max_\theta \sum_N \log \pi_\theta(a_i|s_i)$. The behavior cloning approach works effectively in cases when the demonstrations abound (Ross et al., 2011). However, as robot demonstrations can be costly and time-consuming, we are in favor of a method that can learn from a handful of demonstrations. GAIL (Ho & Ermon, 2016) makes a more efficient use of demonstration data by allowing the agent to interact with the environment and learn from its own experiences. Similar to Generative Adversarial Networks (Goodfellow et al., 2014), GAIL has two networks, a policy network $\pi_\theta : \mathcal{S} \to \mathcal{A}$ and a discriminator network $D_\psi : \mathcal{S} \times \mathcal{A} \to [0, 1]$. GAIL uses a similar min-max objective function as in GANs:

$$\min_\theta \max_\psi \mathbb{E}_{\pi_E}[\log D_\psi(s, a)] + \mathbb{E}_{\pi_\theta}[\log(1 - D_\psi(s, a))], \tag{1}$$

where $\pi_E$ denotes the expert policy that generated the demonstration trajectories. This learning objective encourages the policy $\pi_\theta$ to have an occupancy measure close to the expert policy $\pi_E$.

In practice, we train $\pi_\theta$ with policy gradient methods to maximize the discounted sum of the reward function $r_{gail}(s_t, a_t) = -\log(1 - D_\psi(s_t, a_t))$, clipped by a max value of 10. In continuous domains, trust region methods greatly stabilize policy training. The original GAIL model uses TRPO (Schulman et al., 2015) in the policy update steps. Recently, PPO (Schulman et al., 2017) is proposed as a simple and scalable approximation to TRPO. PPO only relies on the first-order gradients and can be easily implemented with recurrent networks in a distributed setting (Heess et al., 2017). The key idea of PPO is to use the Kullback-Leibler (KL) divergence to dynamically change the coefficient of a regularization term, where the coefficient is adapted based on whether the previous policy update step violates the KL constraint. We use distributed PPO to perform data collection and synchronous gradient updates across many workers in parallel. We trained all our policies with 256 CPU workers, which brings a significant speedup in wall-clock time.

## 3.2 REINFORCEMENT AND IMITATION LEARNING MODEL

### 3.2.1 HYBRID IL/RL REWARD

A common approach to guiding exploration is by engineering a shaping reward. Although reward shaping sometimes provides informative guidance for policy search, it has the well-known drawback of producing suboptimal behaviors (Ng et al., 1999). Hence, we use sparse piecewise constant rewards in this work. Training agents in continuous domains under sparse rewards is particularly challenging. Inspired by reward augmentation introduced in Li et al. 2017 and Merel et al. 2017, we

design a hybrid reward function to mix the imitation reward $r_{gail}$ with the sparse task reward $r_{task}$:

$$r(s_t, a_t) = \lambda r_{gail}(s_t, a_t) + (1 - \lambda)r_{task}(s_t, a_t) \quad \lambda \in [0, 1].  \tag{2}$$

Maximizing this hybrid reward can be interpreted as simultaneous reinforcement and imitation learning, where the imitation reward encourages the policy to generate trajectories closer to demonstration trajectories, and the task reward encourages the policy to achieve high returns in the task. Setting $\lambda$ to either 0 or 1 reduces this method to the standard RL or GAIL setups. Our experiments suggest that with a balanced contribution of these two rewards, the agents can solve tasks that neither GAIL nor RL can solve alone. Further, the final agents achieved higher returns than the human demonstrations owing to the exposure to task rewards.

### 3.2.2 LEVERAGING PHYSICAL STATES IN SIMULATION

The use of simulated system provides us access to the underlying physical states. Even though such privileged information is unavailable on a real system, we can take advantage of it when training the policy in simulation. We propose four techniques to leverage the physical states in simulation to stabilize and accelerate learning, including demonstration curriculum, learning value from states, building object-centric discriminator, and auxiliary tasks.

**Demonstration as a curriculum.** The problem of exploration in continuous domains is exacerbated by the long duration of realistic tasks. Previous work indicates that shaping the distribution of start states towards states where the optimal policy tends to visit can greatly improve policy learning (Kakade & Langford, 2002; Popov et al., 2017). We alter the start state distribution with demonstration states. We build a curriculum that contains clusters of states in different stages of a task. For instance, we define three clusters for the pouring task, including *reaching the mug*, *grasping the mug*, and *pouring*. For a training episode, with probability $\epsilon$ we start it from a random initial state, and with probability $1 - \epsilon$ we uniformly select a cluster and reset the episode to a demonstration state from the cluster. This is possible as our simulated system is fully characterized by the physical states.

**Learning value functions from states.** PPO uses a learnable value function $V_\phi$ to estimate the advantage for policy gradient. During training, each PPO worker executes the policy for $K$ steps and uses the discounted sum of rewards and the value as an advantage function estimator $\hat{A}_t = \sum_{i=1}^{K} \gamma^{i-1} r_{t+i} + \gamma^{K-1} V_\phi(s_{t+K}) - V_\phi(s_t)$, where $\gamma$ is the discount factor. As the policy gradient relies on the value function to reduce variance, it is beneficial to accelerate learning of the value function. Rather than using pixel inputs as the policy network, we take advantage of the low-level physical states (e.g., the position and velocity of the 3D objects and the robot arm) to train the value $V_\phi$ with a smaller multilayer perceptron. We find that training the policy and value in two different modalities stabilizes training and reduces oscillation in the agent's performance. This technique has also been adopted by a concurrent work by Pinto et al. 2017.

**Object-centric discriminator.** Similar to the value function, the GAIL discriminator leverages the physical states to construct task-specific features as its input. In manipulation tasks, we find that object-centric representations (e.g., absolute and relative positions of the objects) provide the salient and relevant signals to the discriminator. The states of the robot arm, on the contrary, tend to make the discriminator too strong and stagnate the training of the policy. Inspired by information hiding strategies used in locomotion domains (Heess et al., 2016; Merel et al., 2017), our discriminator only takes the object-centric features as input while masking out arm-related information.

**State prediction auxiliary tasks.** Auxiliary tasks have been shown effective in improving the learning efficiency and the final performance of deep RL methods (Jaderberg et al., 2016). To facilitate learning visuomotor policies, we add a state prediction layer on the top of the CNN module to predict the locations of objects from the camera observation. We use a fully-connected layer to regress the 3D coordinates of objects in the task. We train this auxiliary task by minimizing the $\ell_2$ loss between the predicted and ground-truth object locations.

### 3.2.3 SIM2REAL POLICY TRANSFER

Policy transfer is shown to a real-world Kinova Jaco robot arm. The simulation was manually aligned to generally match the visuals and dynamics: a Kinect camera was visually calibrated to

match the position and orientation of the simulated camera, and the simulation's dynamics parameters were manually adjusted to match the dynamics of the real arm. Instead of using professional calibration equipment, our approach to sim2real policy transfer relies on domain randomization of camera position and orientation (Tobin et al., 2017; James et al., 2017). in contrast, we do not create intermediate position goals using object position information in reality, but rather train an end-to-end, pixels to velocities, feedback control policy. In addition, to alleviate the issues caused by latency on the real robot, we also fine-tune our policies while subjecting them to action dropping. Detailed descriptions are available in Appendix B.

## 4 Experiments

Here we demonstrate that our proposed approach offers a general framework to visuomotor policy learning. We evaluate the performance of our model in six manipulation tasks illustrated in Fig. 3. We provide more qualitative results in this video.

### 4.1 Environment Setup

We use a Kinova Jaco arm that has 9 degrees of freedom, including six arm joints and three actuated fingers. The robot arm interacts with a diverse set of objects on a tabletop. The visuomotor policy controls the robot using joint velocity commands. Our policy produces 9-dimensional continuous velocities in the range of $[-1, 1]$ at 20Hz. The proprioceptive features consist of the positions and angular velocities of the arm joints and the fingers. We use a positioned camera to collect real-time RGB observations. The proprioceptive features and the camera observations are available in both simulation and real environments. Thus, it enables policy transfer.

We use the MuJoCo physics simulator (Todorov et al., 2012) as our training platform. We use a large variety of objects, from basic geometric shapes to procedurally generated 3D objects as ensembles of primitive shapes. We increase the diversity of objects by randomizing various physical properties, including dimension, color, mass, friction, etc. We used a 3D motion controller called SpaceNavigator, which allows us to operate the robot arm with a position controller, to collect 30 episodes of demonstration for each task and recorded the observations, actions, and physical states into a dataset. As each episode takes less than a minute to complete, demonstrating each task can be done within half an hour.

### 4.2 Robot Arm Manipulation Tasks

Fig. 3 shows visualizations of the six manipulation tasks in our experiments. The first column shows the six tasks in simulated environments, and the second column shows the real-world setup of the block lifting and stacking tasks. We see obvious visual discrepancies of the same task in simulation and reality. These six tasks exhibit learning challenges to varying degrees. The first three tasks use simple colored blocks, which allows us to easily construct the tasks for a real robot. We study sim2real policy transfer with the block lifting and stacking tasks in Sec. 4.4.

**Block lifting.** The goal is to grasp and lift a randomized block, allowing us to evaluate the model's robustness. We vary several random factors, including the robot arm dynamics (friction and armature), lighting conditions, camera poses, background colors, as well as the properties of the block. Each episode starts with a new configuration with these random factors uniformly drawn from a preset range.

**Block stacking.** The goal is to stack one block on top of the other block. Together with the block lifting task, this is evaluated in sim2real transfer experiments.

**Clearing blocks.** This task aims at clearing the tabletop that has two blocks. One strategy to do this task using a single arm is to stack the blocks and pick up both together. This task requires longer time and a more dexterous controller, introducing a significant challenge for exploration.

The next three tasks involve a large variety of procedurally generated 3D shapes, making them difficult to recreate in real environments. We use them to examine the model's ability to generalize across object variations in long and complex tasks.

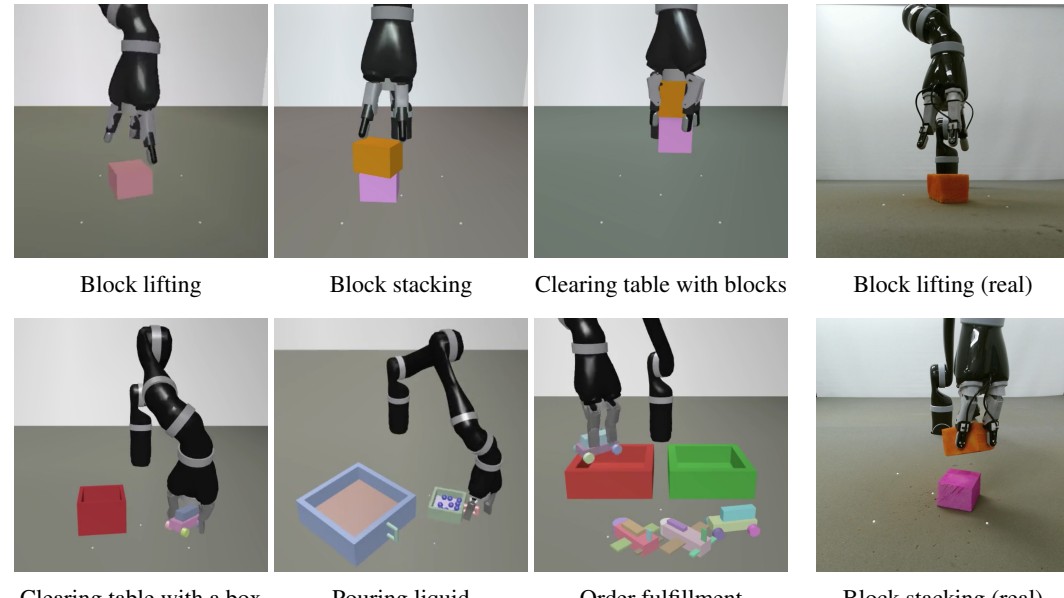

| Block lifting | Block stacking | Clearing table with blocks | Block lifting (real) |

| Clearing table with a box | Pouring liquid | Order fulfillment | Block stacking (real) |

Figure 3: Visualizations of the six manipulation tasks in our experiments. The left column shows RGB images of all six tasks in the simulated environments. These images correspond to the actual pixel observations as input to the visuomotor policies. The right column shows the two tasks with color blocks on the real robot.

**Clearing tabletop.** In this task, the goal is to clear the tabletop that has a box and a toy car. One strategy is to grasp the toy, put it into the box, and lift the box. Both the box and the toy car are randomly generated for each episode.

**Pouring liquid.** Modeling and reasoning about deformable objects and fluids is a long-standing challenge in the robotics community (Schenck & Fox, 2017). We design a pouring task where we use many small spheres to simulate liquid. The goal is to pour the "liquid" from one mug to the other container. This task is particularly challenging due to the dexterity required. Even trained humans struggled to demonstrate the task with our 3D motion controller.

**Order fulfillment.** In this task, we randomly place a variable number of procedurally generated toy planes and cars on the table. The goal is to place all the planes into the green box and all the cars into the red box. This task requires the policy to generalize at an abstract level. It needs to recognize the object categories, perform successful grasps on diverse shapes, and handle tasks with variable lengths.

### 4.3 QUANTITATIVE EVALUATION

Our full model can solve all six tasks, with only occasional failures, using the same policy network, the same training algorithm, and a fixed set of hyperparameters. On the contrary, neither reinforcement nor imitation alone can solve all tasks. We compare the full model with three baseline methods, where we evaluate degenerated versions of our model, which correspond to RL, GAIL, and RL w/o demonstration curriculum. These baselines use the same setup as the full model, except that we set $\lambda = 0$ for RL and $\lambda = 1$ for GAIL, while our model uses a balanced contribution of the hybrid reward, where $\lambda = 0.5$. In the third baseline, all the training episodes start from random initial states rather than resetting to demonstration states. This corresponds to a standard RL setup.

We report the mean episode returns as a function of the number of training iterations in Fig. 4. Our full model achieves the highest returns in all six tasks. The only case where the baseline model is on par with the full model is the block lifting task, in which both the RL baseline and the full model achieved similar levels of performance. We hypothesize that this is due to the short length of the lifting task, where random exploration in RL is likely to reach the goal states without the aid of GAIL.

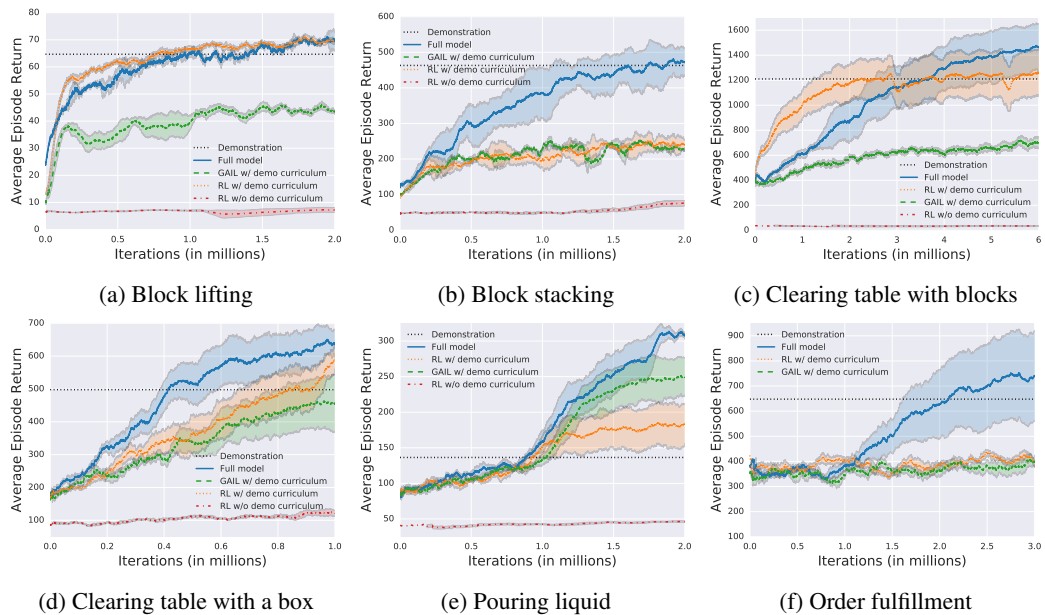

Figure 4: Learning efficiency of our reinforcement and imitation model against baselines. The plots are averaged over 5 runs with different random seeds. All the policies use the same network architecture and the same hyperparameters (except $\lambda$).

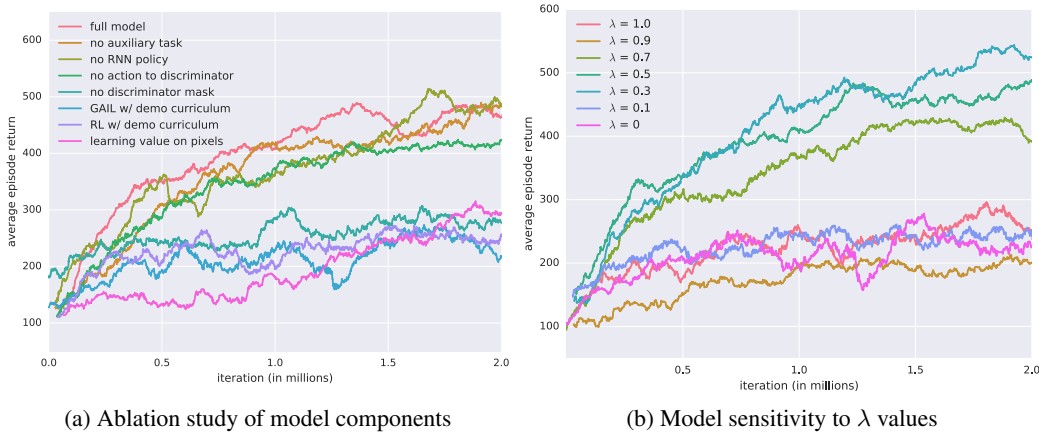

Figure 5: Model analysis in the stacking task. On the left we investigate the impact on performance by removing each individual component from the full model. On the right we investigate the model's sensitivity to the hyperparameter $\lambda$ that moderates the contribution of reinforcement and imitation.

In the other five tasks, the full model outperforms both the reinforcement learning and imitation learning baselines by a large margin, demonstrating the effectiveness of combining reinforcement and imitation for learning complex tasks. Comparing the two variants of RL with and without using demonstration as a curriculum, we see a pronounced effect of altering the start state distribution. We see that RL from scratch leads to a very slow learning progress; while initiating episodes along demonstration trajectories enables the agent to train on states from different stages of a task. As a result, it greatly reduces the burden of exploration and improves the learning efficiency. We also report the mean episode returns of human demonstrations in these figures. While demonstrations using the 3D motion controller are imperfect, especially for pouring (see video), the trained agents can surpass them via interacting with the environment.

Two findings are noteworthy. First, the RL agent learns faster than the full model in the table clearing task, but the full model eventually outperforms. This is because the full model discovers

a novel strategy, different from the strategy demonstrated by human operators (see video). In this case, imitation gave contradictory signals but eventually, reinforcement learning guided the policy towards a better strategy. Second, pouring liquid is the only task where GAIL outperforms its RL counterpart. Imitation can effectively shape the agent's behaviors towards the demonstration trajectories (Wang et al., 2017). This is a viable solution for the pouring task, where a controller that generates similar-looking behaviors can complete the task. In contact-rich domains, however, a controller learned solely from dozens of demonstrations would struggle to handle complex object dynamics and to infer the true task goal. We hypothesize that this is why the baseline RL agent outperforms the GAIL agent in the other five tasks.

We further perform an ablation study in the block stacking task to understand the impacts of different components of our model. In Fig. 5a, we trained our agents with a number of configurations, each with a single modification to the full model. We see that the final performances of the experiments cluster into two groups: agents that learn to stack (with average returns greater than 400) and agents that only learn to lift (with average returns between 200 and 300). These results indicate that the hybrid RL/IL reward, learning value function from states, and object-centric discriminator play an integral role in learning good policies. Using sole RL or GAIL reward, learning value function on pixels, or no information hiding for discriminator input (no discriminator mask) all result in inferior performances. In contrast, the optional components include the recurrent policy core (LSTM), the use of state prediction auxiliary tasks, and whether to include actions in discriminator input. We then examine the model's sensitivity to the $\lambda$ values in Eq. 2. We see in Fig. 5b that, our model works well with a broad range of $\lambda$ values from 0.3 to 0.7 that provide a balanced mix of the RL and GAIL rewards.

### 4.4    SIM2REAL POLICY TRANSFER RESULTS

To assess the robustness of the simulation-trained policy, we evaluate zero-shot transfer (no additional training) on a real Jaco arm. Given a real-world set up that mirrored the simulated domain to a large extent, including camera positions and robot kinematics and approximate object size and color, we ran the trained network policy and counted the number of successful trials for both the lifting and stacking tasks. Although the sim and real domains were similar, there was still a sizable reality gap that made zero-shot transfer challenging. For example, the objects were non-rigid foam blocks which deformed and bounced unpredictably. The arm position was randomly initialized and the target block(s) placed in a number of repeatable start configurations for each task. The zero-shot transfer of the lifting policy had a success rate of 64% over 25 trials (split between 5 block configurations). The stacking policy had a success rate of 35% over 20 trials (split between 2 block configurations). 80% of the stacking trajectories, however, contain successful lifting behavior. Qualitatively, the policies are notably robust even on failed attempts — rather than exhibiting "open-loop" behaviors such as attempting to stack a non-existent block, the policy repeatedly chases the block to get a successful grasp before trying to stack (see video). For more detailed descriptions of the sim2real results, refer to Appendix B.

## 5    CONCLUSION

We have shown that combining reinforcement and imitation learning considerably improves the agents' ability to solve challenging dexterous manipulation tasks from pixels. Our proposed method sheds light on the three stages of a principled pipeline for robot skill learning: first, we collected a small amount of demonstration data to simplify the exploration problem; second, we relied on physical simulation to perform large-scale distributed robot training; and third, we performed sim2real transfer for real-world deployment. In future work, we seek to improve the sample efficiency of the learning method and to leverage real-world experience to close the reality gap for policy transfer.

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

## A  EXPERIMENT DETAILS

The policy network takes the pixel observation and the proprioceptive feature as input. The pixel observation is an RGB image of size $64 \times 64 \times 3$. We used the Kinect for Xbox One camera in the real environment. The proprioceptive feature describes the joint positions and velocities of the Kinova Jaco arm. Each joint position is represented as the $\sin$ and $\cos$ of the angle of the joint in joint coordinates. Each joint velocity is represented as the scalar angular velocity. This results in a 24-dimensional proprioceptive feature that contains the positions (12-d) and velocities (6-d) of the six arm joints and the positions (6-d) of the three fingers. We exclude the finger velocities due to the noisy sensory readings on the real robot.

We used Adam (Kingma & Ba, 2014) to train the neural network parameters. We set the learning rate of policy and value to $10^{-4}$ and $10^{-3}$ respectively, and $10^{-4}$ for both the discriminator and the auxiliary tasks. The pixel observation is encoded by a two-layer convoluational network. We use 2 convolutional layers followed by a fully-connected layer with 128 hidden units. The first convolutional layer has 16 $8 \times 8$ filters with stride 4 and the second 32 $4 \times 4$ filters with stride 2. We add a recurrent layer of 100 LSTM units before the policy and value outputs. The policy output is the mean and the standard deviation of a conditional Gaussian distribution over the 9-dimensional joint velocities. The initial policy standard deviation is set to $\exp(-3)$ for the *clearing table with blocks* task and $\exp(-1)$ for the other five tasks. The auxiliary head of the policy contains a separate 3-layer MLP sitting on top of the convolutional network. The first two layers of the MLP has 200 and 100 hidden units respectively, while the third layer predicts the auxiliary outputs. Finally, the discriminator is a simple three-layer MLP of 100 and 64 hidden units for the first two layers with the third layer producing log probabilities. The networks use $\tanh$ nonlinearities.

We trained the visuomotor policies using the distributed PPO algorithm (Heess et al., 2017) with synchronous gradient updates from 256 CPU workers. Each worker runs the policy to complete an entire episode before the parameter updates are computed. We set a constant episode length for each task based on its difficulty, with the longest being 1000 time steps (50 seconds) for the *clearing table with blocks* and *order fulfillment* tasks. We set $K = 50$ as the number of time steps for computing $K$-step returns and truncated backpropagation through time to train the LSTM units. After a worker collects a batch of data points, it performs 50 parameter updates for the policy and value networks, 5 for the discriminator and 5 for the auxiliary prediction network.

## B  SIM2REAL DETAILS

To better facilitate sim2real transfer, we lower the frequency at which we sample the observations. Pixel observations are only observed at the rate of 5Hz despite the fact that our controller runs at 20Hz. Similarly, the proprioceptive features are observed at a rate of 10Hz. In addition to observation delays, we also apply domain variations. Gaussian noise (of standard deviation 0.01) are added proprioceptive features. Uniform integers noise in the range of $[-5, 5]$ are added to each pixel independently. Pixels of values outside the range of $[0, 255]$ are clipped. We also vary randomly the shade of grey on the Jaco arm, the color of the table top, as well as the location and orientation of the light source (see Fig. 6).

In the case of block lifting, we vary in addition the dynamics of the arm. Specifically, we dynamically change the friction, damping, armature, and gain parameters of the robot arm in simulation to further robustify the agent's performance.

### B.1  ACTION DROPPING

Our analysis indicates that, on the real robot, there is often a delay in the execution of actions. The amount of delay also varies significantly. This has an adverse effect on the performance of our agent on the physical robot since our agents' performance depends on the timely execution of their actions. To better facilitate the transfer to the real robot, we fine-tune our trained agent in simulation while subjecting them to a random chance of dropping actions. Specifically, each action emitted by the agent has a $50\%$ chance of being executed immediately in which case the action is flagged as the last executed action. If the current action is not executed, the last executed action will then be executed.

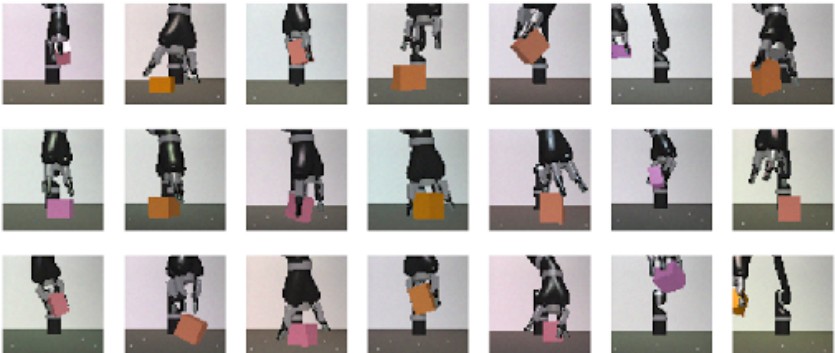

Figure 6: Tiles show the representative range of diversity seen in the domain-randomized variations of the colors, lighting, background, etc.

Table 1: Block Lifting success rate from different positions. (**LL, LR, UL, UR, and C** represents the positions of lower left, lower right, upper left, upper right, and center respectively.)

|  | LL | LR | UL | UR | C | All |
|---|---|---|---|---|---|---|
| No Action Dropping | 2/5 | 2/5 | 1/5 | 3/5 | 4/5 | 12/25 |
| Action Dropping | 4/5 | 4/5 | 4/5 | 0/5 | 4/5 | 16/25 |

Using the above procedure, we fine-tune our agents on both block lifting and block stacking for a further 2 million iterations.

To demonstrate the effectiveness of action dropping, we compare our agent on the real robot over the task of block lifting. Without action dropping, the baseline agent lifts $48\%$ percent of the time. After fine-tuning using action dropping, our agent succeeded $64\%$ percent of the time. For the complete set of results, please see Table 1 and Table 2.

## C   TASK DETAILS

We use a fixed episode length for each task, which is determined by the amount of time a skilled human demonstrator can complete the task. An episode terminates when a maximum number of agent steps are performed. The robot arm operates at a control frequency of 20Hz, which means each time step takes 0.05 second.

We segment into a sequence of stages that represent an agent's progress in a task. For instance, the *block stacking* task can be characterized by three stages, including *reaching the block*, *lifting the block* and *stacking the block*. We define functions on the underlying physical state to determine the stage of a state. This way, we can cluster demonstration states according to their corresponding stages. These clusters are used to reset training episodes in our demonstration as a curriculum technique proposed in Sec. 3.2.2. The definition of stages also gives rise to a convenient way of specifying the reward functions without hand-engineering a shaping reward. We define a piecewise constant reward function for each task, where we assign the same constant reward to all the states that belong to the same stage. We detail the stages, reward functions, auxiliary tasks, and object-centric features for the six tasks in our experiments.

**Block lifting.** Each episode lasts 100 time steps. We define three stages and their rewards (in parentheses) to be initial (0), reaching the block (0.125) and lifting the block (1.0). The auxiliary task is to predict the 3D coordinates of the color block. The object-centric feature consists of the relative position between the gripper and the block.

**Block stacking.** Each episode lasts 500 time steps. We define four stages and their rewards to be initial (0), reaching the orange block (0.125), lifting the orange block (0.25), and stacking the orange block onto the pink block (1.0). The auxiliary task is to predict the 3D coordinates of the two blocks.

Table 2: Success rate of the block stacking agent from different starting positions. **Left** and **Right** indicates the position of the support block upon initialization.

|  | Left | Right | All |
|---|---|---|---|
| Stacking Success Rate | 5/10 | 2/10 | 7/20 |
| Lifting Success Rate | 9/10 | 7/10 | 16/20 |

The object-centric feature consists of the relative positions between the gripper and the two blocks respectively.

**Clearing table with blocks.** Each episode lasts 1000 time steps. We define five stages and their rewards to be initial (0), reaching the orange block (0.125), lifting the orange block (0.25), stacking the orange block onto the pink block (1.0), and lifting both blocks off the ground (2.0). The auxiliary task is to predict the 3D coordinates of the two blocks. The object-centric feature consists of the 3D positions of the two blocks as well as the relative positions between the gripper and the two blocks respectively.

**Clearing table with a box.** Each episode lasts 500 time steps. We define five stages and their rewards to be initial (0), reaching the toy (0.125), grasping the toy (0.25), putting the toy into the box (1.0), and lifting the box (2.0). The auxiliary task is to predict the 3D coordinates of the toy and the box. The object-centric feature consists of the 3D positions of the toy and the box as well as the relative positions between the gripper and these two objects respectively.

**Pouring liquid.** Each episode lasts 500 time steps. We define three stages and their rewards to be initial (0), grasping the mug (0.05), pouring ($0.1N$), where $N$ is the number of small spheres in the other container. The auxiliary task is to predict the 3D coordinates of the mug. The object-centric feature consists of the 3D positions of the mug, the relative position between the gripper and the mug, and the relative position between the mug and the container.

**Order fulfillment.** Each episode lasts 1000 time steps. The number of objects varies from 1 to 4 across episodes. We define five stages that correspond to the number of toys in the boxes. The immediate reward corresponds to the number of toys placed in the correct boxes (number of toy planes in the green box and toy cars in the red box). To handle the variable number of objects, we only represent the objects nearest to the gripper for the auxiliary task and the object-centric feature. The auxiliary task is to predict the 3D coordinates of the nearest plane and the nearest car to the gripper. The object-centric feature consists of the relative positions from the gripper to these two nearest objects.

