# OpenReview forum: "Reinforcement and Imitation Learning for Diverse Visuomotor Skills"
_ICLR.cc/2018/Conference — Reject_

### Official Review · AnonReviewer3 · 2017-11-25
**Good experimental work, but some pretty severe over-statements and claims**

**Rating:** 6
**Confidence:** 5

**Review:**

Paper summary: The authors propose a number of tricks to enable training policies for pick and place style tasks using a combination of GAIL-based imitation learning and hand-specified rewards, as well as use of unobserved state information during training and hand-designed curricula. The results demonstrate manipulation policies for stacking blocks and moving objects, as well as preliminary results for zero-shot transfer from simulation to a real robot for a picking task and an attempt at a stacking task.

Review summary: The paper proposes a limited but interesting contribution that will be especially of interest to practitioners, but the scope of the contribution is somewhat incremental in light of recent work, and the results, while interesting, could certainly be better. In the balance, I think the paper should be accepted, because it will be of value to practitioners, and I appreciate the detail and real-world experiments. However, some of the claims should be revised to better reflect what the paper actually accomplishes: the contribution is a bit limited in places, but that's *OK* -- the authors should just be up-front about it.

Pros:
- Interesting tasks that combine imitation and reinforcement in a logical (but somewhat heuristic) way
- Good simulated results on a variety of pick-and-place style problems
- Some initial attempt at real-world transfer that seems promising, but limited
- Related work is very detailed and I think many will find it to be a very valuable overview
Cons:
- Some of the claims (detailed below) are a bit excessive in my opinion
- The paper would be better if it was scoped more narrowly
- Contribution is a bit incremental and somewhat heuristic
- The experimental results are difficult to interpret in simulation
- The real-world experimental results are not great
- There are a couple of missing citations (but overall related work is great)

Detailed discussion of potential issues and constructive feedback:

> "Our approach leverages demonstration data to assist a reinforcement learning agent in learning to solve a wide range of tasks, mainly previously unsolved."
>> This claim is a bit peculiar. Picking up and placing objects is certainly not "unsolved," there are many examples. If you want image-based pick and place with demonstrations for example, see Chebotar '17 (not cited). If you want stacking blocks, see Nair '17. While it's true that there is a particular combination of factors that doesn't exactly appear in prior work, the statement the authors make is way too strong. Chebotar '17 shows picking and placing a real-world objective with a much higher success rate than reported here, without simulation. Nair '17 shows a much harder stacking task, but without images -- would that method have worked just as well with image-based distillation? Very likely. Rajeswaran '17 shows tasks that arguably are much harder. Maybe a more honest statement is that this paper proposes some tasks that prior methods don't show, and some prior methods show tasks that the proposed method can't solve. But as-is, this statement misrepresents prior work.

> Previous RL-based robot manipulation policies (Nair et al., 2017; Popov et al., 2017) largely rely on low-level states as input, or use severely limited action spaces that ignore the arm and instead learn Cartesian control of a simple gripper. This limits the ability of these methods to represent and solve more complex tasks (e.g., manipulating arbitrary 3D objects) and to deploy in real environments where the privileged state information is unavailable.
>> This is a funny statement. Some use images, some don't. There is a ton of prior work on RL-based robot manipulation that does use images. The current paper does use object state information during training, which some prior works manage to avoid. The comments about Cartesian control are a bit peculiar... the proposed method controls fingers, but the hand is simple. Some prior works have simpler grippers (e.g., Nair) and some have much more complex hands (e.g., Rajeswaran). So this one falls somewhere in the middle. That's fine, but again, this statement overclaims a bit.

> To sidestep the constraints of training on real hardware we embrace the sim2real paradigm which
has recently shown promising results (James et al., 2017; Rusu et al., 2016a).
>> Probably should cite Sadeghi et al. and Tobin et al. in regard to randomization, both of which precede James '17.

> we can, during training, exploit privileged information about the true system state
>> This was done also in Pinto et al. and many of the cited GPS papers

> our policies solve the tasks that the state-of-the-art reinforcement and imitation learning cannot solve
>> I don't think this statement is justified without much wider comparisons -- the authors don't attempt any comparisons to prior work, such as Chebotar '17 (which arguably is closest in terms of demonstrated behaviors), Nair '17 (which is also close but doesn't use images, though it likely could).

> An alternative strategy for dealing with the data demand is to train in simulation and transfer
>> Aside from previously mentioned citations, should probably cite Devin "Towards Adapting Deep Visuomotor Representations"

> Sec 3.2.1
>> This method seems a bit heuristic. It's logical, but can you say anything about what this will converge to? GAIL will try to match the demonstration distribution, and RL will try to maximize expected reward. What will this method do?

> Experiments
>> Would it be possible to indicate some measure of success rate for the simulated experiments? As-is, it's hard to tell how well either the proposed method or the baselines actually work.

> Transfer
>> My reading of the transfer experiments is that they are basically unsuccessful. Picking up a rectangular object with 80% success rate is not very good. The stacking success rate is too low to be useful. I do appreciate the authors trying out their method on a real robotic platform, but perhaps the more honest assessment of the outcome of these experiments is that the approach didn't work very well, and more research is needed. Again, it's *OK* to say this! Part of the purpose of publishing a paper is to stimulate future research directions. I think the transfer experiments should definitely be kept, but the authors should discuss the limitations to help future work address them, and present the transfer appropriately in the intro.

> Diverse Visuomotor Skills
>> I think this is a peculiar thing to put in the title. Is the implication that prior work is not diverse? Arguably several prior papers show substantially more diverse skills. It seems that all the skills here are essentially pick and place skills, which is fine (these are interesting skills), but the title seems like a peculiar jab at prior work not being "diverse" enough, which is simply misleading.

---

> ### Comment · AnonReviewer3 · 2018-01-12
> **Rebuttal response**
>
> I’ve read the author response and maintain my score of the paper.
>
> I will add that I find the author response quite disappointing. It seems that instead of toning down the claims in their paper, the authors instead chose to double down by stating that
>
> > it is only until recently that model-free deep RL models achieved some initial success in robotic manipulation tasks
>
> I would argue that this statement is incorrect. As discussed at length in my review, prior work has demonstrated a range of robotic manipulation skills, many more successful than those in this paper, and some without the use of simulation or without demonstrations. This is not just true for the relatively recent Rajeswaran and Nair papers, but also James, Chebotar, and many many others. I agree that the particular experimental details matter a great deal, but the authors seem to want to have it both ways: dismiss the importance of additional assumptions and downsides of their method (demonstrations, an accurate simulator, etc) and emphasize the assumptions in prior work. As I wrote before, I don’t think the contribution is below bar, but the claims are in my opinion quite excessive.

---

> > ### Author Response · Authors · 2018-01-29
> > **Reply**
> >
> > We thank the reviewer for agreeing that "the particular experimental details matter a great deal" when it comes to the technical contributions. We will clearly circumscribe our contribution claims in the context of previous work in our next revision.

---

### Official Review · AnonReviewer2 · 2017-11-26
**Nice results, but approach only works in simulation**

**Rating:** 4
**Confidence:** 4

**Review:**

This paper claims to present a "general deep reinforcement learning" method that addresses the issues of real-world robotics: data constraints, safety, and lack of state information, and exploration by using demonstrations. However, this paper actually addresses these problems by training in a simulator, and only transferring 2 of the 6 tasks to the real world. The real world results are  lackluster. However, the simulated results are nice.

The method in the paper is as follows: the environment reward is augmented by a reward function learned from human demonstrations using GAIL on full state (except for the arm). Then, an actor-critic method is used where the critic gets full state information, while the actor needs to learn from an image. However, the actor's convolutional layers are additionally trained to detect the object positions.

Strengths:
+ The simulated tasks are novel and difficult (sorting, clearing a table)
+ Resetting to demonstration states is a nice way to provide curriculum

Limitations:
+ The results make me worries that the simulation environments have been hyper-tailored to the method, as the real environments looks very similar, and should transfer.
+ Each part of the method is not particularly novel. Combining IRL and RL has been done before (as the authors point out in the related work), side-training perception module to predict full state has been done before ("End-to-end visuomotor learning"), diversity of training conditions has been done before (Domain randomization).
+ Requiring hand-specified clusters of states for both selecting starting states and defining a reward functions requires domain knowledge. Why can't they be clustered using a clustering method?
+ Because the method needs simulation to learn a policy, it is limited to tasks that can be simulated somewhat accurately (e.g. ones with simple dynamics). As shown by the poor transfer of the stacking task, block stacking with foam blocks is not a such task.


Questions:
+ How many demonstrations do you use per task?
+ What are the "relative" positions included in the "object-centric" state input?

Misleading parts of the paper:
+ The introduction of the paper primes the reader to expect a method that can work on a real system. However, this method only gets 64% accuracy on a simple block lifting task, 35% on a stacking task.
+ "Appendix C. "We define functions on the underlying physical state to determine the stage of a state…The definition of stages also gives rise to a convenient way of specifying the reward functions without hand-engineering a shaping reward. "-> You are literally hand engineering a shaping reward. The main text misleadingly refers to "sparse reward", which usually refers to a single reward upon task completion.

In conclusion, I find that the work lacks significance because the results are dependent on a list of hacks that are only possible in simulation.

---

> ### Comment · AnonReviewer2 · 2018-01-12
> **Revision of review**
>
> I agree with the authors that they should not be judged against Nair 17, Rajeswaran 17, and Chebotar 17, as those are concurrent papers that are probably under review at this time.
>
> However, the author response has not addressed what I consider to be major limitations to this paper. These are:
> + The algorithm itself requires simulation. As the authors themselves point out:  “dynamics mismatch between the simulated physical engine and real systems, e.g., simulated rigid body v.s. soft foam, introduces another major challenge.”
> + Requiring hand-specified clusters of states for both selecting starting states and defining a reward functions requires domain knowledge.
>
> As the authors point out, “Our goal of applying deep RL to robotic manipulation is *not* to find one solution that can solve a particular instance of tasks, e.g., block stacking, with a 100% success rate. As a matter of fact, the latest video of the backflipping robot from Boston Dynamics has demonstrated how far we can go with an hand-engineered solution.”
>
> Unfortunately, I find that there solution is significantly more hand engineered than they claim. The hand-specified clusters require domain knowledge for each task. The “object centric features” differ between tasks: while some tasks just use all pairwise relative positions between objects, the “pouring” task only uses the relative position between the gripper and the mug, and the relative position between the mug and the container.
> ——>>More surprisingly, for the plane/car sorting task, only the NEAREST plane and car are included in the features! This is hidden away in the last section of the appendix.
>
> In sum, this paper uses a lot of hand-tweaked representations and rewards in order to obtain impressive looking simulation results. While it is good to get these results at all, I do not think that this is a good fit for ICLR. The method and results would be better at a robotics conference. However, I also find that the method/results were not presented in good faith, and were often misleading or overstated. My review rating remains a 4.

---

> > ### Author Response · Authors · 2018-01-29
> > **Reply**
> >
> > We thank this reviewer for the additional feedback. We would like to address the reviewer’s comments on the use of simulation and the amount of hand engineering in this work. We will also make an effort to clearly describe our engineering components in the next version of the draft.
> >
> > We acknowledged that simulation is at the center of our approach. The use of simulation is a design choice due to the high sample complexity of today’s model-free deep RL methods as well as practical concerns such as safety and cost. Admittedly, the value of simulation is hampered by the model mismatch between sim and real. This requires better sim2real techniques to minimize the domain gap. Sim2real is currently an active field of research, which is not yet solved. Our effort to perform real-world experiments is to exemplify how zero-shot sim2real transfer can achieve some initial success, or in other words, fail less than expected on the real robot. Our results demonstrated that it is hopeful to learn end-to-end control from raw pixel inputs to velocity commands with the aid of simulation. Furthermore, this zero-shot transfer experiments implied the possibility of leveraging real experiences to improve the controllers.
> >
> > RL training assumes a reward function, which is generally unavailable or at least hard to obtain on real hardware. The use of simulation permitted us to define the reward functions and object-centric features, which are crucial for policy training. For training in simulation, it is a common practice to leverage the simulator states and domain knowledge to define the rewards (see Nair 17, Rajeswaran 17). For long-horizon tasks, a binary task-completion reward is often insufficient for RL random exploration (see Popov 17). Our reward functions are sparse piecewise constant functions that correspond to different stages of a task. A similar “step” reward has been used in Nair 17 to enable solving longer horizon tasks (e.g., stacking more than 4 blocks). Empirically, we found that defining such multi-stage rewards is easier than designing a dense shaping reward and less prone to converging on suboptimal behaviors. In terms of the object-centric features, these features are important for the GAIL discriminator to concentrate on the features that are relevant to the task goal rather than discriminating based on spurious information. Our preliminary experiments found that our model is not sensitive to the particular choice of object-centric features. As long as these features carry enough task-specific information, the GAIL discriminator is able to provide supervision signal for successful training.
> >
> > Finally, the exploitation of privileged and task-specific information is only required for training. We ultimately produced vision-based policy that does not rely on such hand-engineered features.

---

### Official Review · AnonReviewer1 · 2017-11-27
**Minor contributions leading to relatively poor results on real robot**

**Rating:** 4
**Confidence:** 4

**Review:**

Given that imitation learning is often used to initialize reinforcement learning, the authors should consider using a more descriptive title for this paper.

The main contribution of the paper is to use a mixture of the reinforcement learning reward and the imitation learning signal from GAIL. This approach is generally fairly straightforward, but seems to be effective in simulation, and boils down to equation 2. It would be interesting to discuss and evaluate the effects of changing lambda overtime.

The second contribution can be seen as a list of four techniques for getting the most out of using the simulation environment and the state information that it provides. A list of these types of techniques could be useful for students training networks on simulations, although each point on the list is fairly straightforward. This part also leads to an ablation study to determine the effects of the proposed techniques. The results plot should include error bars.

The earlier parts of the experiment were evaluated on three additional tasks. Although these tasks are all variations of putting things into a box, they do add some variability to the experiments. It was also great seeing the robot learning multiple strategies for the table clearing task.

The third part is transferring the simulation policy to the real robot. This section and the additional supplementary material are fairly short and should be expanded upon. It seems as though the transfer mainly depends on learning from randomized domains to achieve more robust policies that can then be applied to the real domain. The transfer learning is a crucial step of the pipeline. Unfortunately the results are not great. A 64% success rate for lifting the block and 35% for stacking are both fairly low. The lifting success is somewhat higher for the stacking at 80% with repeated attempts. The authors need to discuss these results. What is the cause of these low success rates? Is it the transfer learning or due to an earlier step in the pipeline? How do these success rates change with the variance in the training scenarios?

How much of the shape variability is being accounted for by the natural adaptability of the hand? If you give the robot images with one set of objects, but the actual task is performed  using objects of different shapes and sizes, how much does the performance decrease?

---

### Author Response · Authors · 2017-12-30
**Our response**

We agree with the reviewers that some revision is necessary for clarifying the claims and better summarizing our contributions in presence of previous and concurrent work. We will tone down the introduction and provide more backups for our claims in the next version of the draft.

However, we would like to point out that it is only until recently that model-free deep RL models achieved some initial success in robotic manipulation tasks. Several *concurrent* works, which are cited in our draft, have explored similar techniques used in our model. [Nair et al. '17, arxiv 28 Sep 2017] and [Rajeswaran et al. '17, arxiv 28 Sep 2017] have leveraged demonstrations in reinforcement learning; [Pinto et al. '17, arxiv 18 Oct 2017] have also low-level simulation states to train the critic. [Peng et al. '17, arxiv 18 Oct 2017] randomized system dynamics to achieve sim2real transfer. Note that, all these works are released on arxiv less than one month before the ICLR deadline, and none is thus far accepted in a peer-reviewed conference. We developed our approach completely independently. The concurrent works each solved one small piece of a big puzzle. Our model integrates the techniques used in these works along with a series of novel features into a single approach. As a result, it can solve more challenging robotic control problems than what have been demonstrated in these works. Therefore, we believe that the contributions of our work should not be diminished in presence of these concurrent works.

Also, while similar tasks may have been studied in the literature, seemingly similar tasks can exhibit quite different degrees of complexity based on a combination of factors, such as the controller (position [Pinto et al. '17] vs 9-DoF joint velocity [Ours]), the input modalities (states [Nair et al. '17] v.s. pixels [Ours]), variations of initial configurations and objects (a fixed set of objects [Rajeswaran et al. '17] vs procedurally generated objects [Ours]), etc. Even for the same task, different training protocols can have a major impact on the complexity of the task as well the generality and flexibility of the approach, such as behavior cloning [James et al. '17] v.s. reinforcement learning [Ours], pretrained vision module [Chebotar et al. '17] v.s. end-to-end learning [Ours]).

A simple thought experiment that we can use to estimate the complexity of a task is to imagine the efforts required to hand-engineer a controller that solves the task. In the case of block stacking, it seems feasible to design a scripted policy to stack a set of fixed-sized blocks with Cartesian control given low-level state information [Nair et al. '17]. Indeed, such scripted controllers have been used e.g. in [Duan et al. '17]. However, it is significantly more demanding, if not intractable, to write such a controller to stack blocks of different sizes with a 9-DoF joint velocity controller given camera inputs [Ours].

Our goal of applying deep RL to robotic manipulation is *not* to find one solution that can solve a particular instance of tasks, e.g., block stacking, with a 100% success rate. As a matter of fact, the latest video of the backflipping robot from Boston Dynamics has demonstrated how far we can go with an hand-engineered solution. Our goal is to derive a flexible approach that can be applied to a wide variety of tasks. To this end, in the paper we show that the same techniques and network can be used to solve a wide range of tasks, including stacking and lifting, pouring, etc.

Our sim2real experiments focused on zero-shot policy transfer. This requires us to minimize the domain gap between the simulator and the real system. As shown in Fig. 3, even after careful tuning, the discrepancy between simulation rendering and the real camera frame is still apparent. Neural network policies can be sensitive to such nuances. This caused a major challenge for sim2real transfer. Furthermore, dynamics mismatch between the simulated physical engine and real systems, e.g., simulated rigid body v.s. soft foam, introduces another major challenge. Future research is required to enable better policy transfer, potentially by leveraging a small amount of real-world experiences on the real hardware. The state-of-the-art work closest to our setup is [Rusu et al. '17], which used progressive networks to train a pixel-to-action RL policies for real robots. It has been only demonstrated in a block reaching task. In comparison, our sim2real block stacking agent can perfectly solve the block reaching subtask at a 100% success rate.

---

> ### Author Response · Authors · 2017-12-30
> **Answers to some minor questions**
>
> + How many demonstrations do you use per task?
> We used 30 demonstrations for each task, which can be collected within half an hour, described in Sec. 4.1.
>
> + What are the "relative" positions included in the "object-centric" state input?
> The relative positions include the difference between (x,y,z) coordinates of the objects and the robot gripper. The details are described in the appendix Sec. C.

---

### Decision · Program_Chairs · 2018-01-29
**ICLR 2018 Conference Acceptance Decision**

**Decision:**

Reject

**Comment:**

While the reviewers agree that this paper does provide a contribution, it is small and does overlap with several concurrent works. it is a bit hand-engineered.
The authors have provided a lengthy rebuttal, but the final reviews are not strong enough.